



# Observations of precipitation energies during different types of pulsating aurora

Fasil Tesema[1,2], Noora Partamies[1,2], Hilde Nesse Tyssøy[2], and Derek McKay[3]

[1]The University Centre in Svalbard (UNIS), Norway,
[2]Birkeland Centre for Space Science, University of Bergen, Norway,
[3]NORCE Norwegian Research Centre AS, Tromsø, Norway.

**Correspondence:** Fasil Tesema (fasil.tesema@unis.no)

**Abstract.** Pulsating aurora (PsA) is a diffuse type of aurora with different structures switching on and off with a period of few seconds. It is often associated with energetic electron precipitation ($>10\,$keV) resulted in the interaction between magnetospheric electrons and electromagnetic waves in the magnetosphere. Recent studies categorize pulsating aurora into three different types: amorphous pulsating aurora (APA), patchy pulsating aurora (PPA), and patchy aurora (PA) based on the spatial extent of pulsations and structural stability. Differences in precipitation energies of electrons associated with these types of pulsating aurora have been suggested. In this study, we further examine these three types of pulsating aurora using electron density measurements from the European Incoherent Scatter (EISCAT) VHF/UHF radar experiments and Kilpisjärvi Atmospheric Imaging Receiver Array (KAIRA) cosmic noise absorption (CNA) measurements. Based on ground-based all-sky camera images over the Fennoscandian region, we identified a total of 92 PsA events in the years between 2010 and 2020 with simultaneous EISCAT experiments. Among these events, 39, 35, and 18 were APA, PPA, and PA types with a collective duration of 58 hrs, 43 hrs, and 21 hrs, respectively. We found that below 100 km, electron density enhancements during PPAs and PAs are significantly higher than during APA. However, there are no appreciable electron density differences between PPA and APA above 100 km, while PA showed weaker ionization. The altitude of the maximum electron density also showed considerable differences among the three types, centered around 110 km, 105 km, and 105 km for APA, PPA, and PA, respectively. The KAIRA CNA values also showed higher values on average during PPA (0.33 dB) compared to PA (0.23 dB) and especially APA (0.17 dB). In general, this suggests that the precipitating electrons responsible for APA have a lower energy range compared to PPA and PA types. Among the three categories, the magnitude of the maximum electron density shows higher values during PPA at lower altitudes and in the late MLT sector (after 5 MLT). We also found significant ionization down to 70 km during PPA and PA, which corresponds to $\sim$200 keV energies of precipitating pulsating aurora electrons.

*Copyright statement.* TEXT





# 1 Introduction

The interaction between the solar wind and the magnetosphere results in particle precipitation into the Earth's atmosphere through many different processes. Particles from the plasma sheet and radiation belts are accelerated and scattered into the loss cone to eventually collide with the species in the Earth's polar atmosphere. These collisions cause the atmospheric gas to glow in different shimmering bands of color in the sky, called aurora. The most common colors of the aurora are blue, green, and red at the wavelength of 427.8 nm, 557.7 nm, and 630.0 nm, respectively. However, an auroral spectrum ranges from ultraviolet to infra-red wavelengths depending on the type of atmospheric gas that undergoes emission. In general, the electrons generating aurora have energies ranging from 100 eV to 100 keV, which affects the atmosphere by ionizing and changing the chemistry. The auroras are varied in appearance they displayed in the sky due to different magnetospheric processes; most are visible as discrete auroras with ribbons, arcs, and spirals, and some are visible as blinking patches of light called pulsating auroras (PsA).

Pulsating auroras are mostly characterized as quasi-periodic low-intensity (few kilo Rayleigh) diffuse emission, which switches on and off with periods of a few seconds to a few tens of seconds (Royrvik and Davis, 1977; Yamamoto et al., 1988). The structures of PsA can be irregularly shaped patches, or thin arcs elongated in the east-west direction (Wahlund et al., 1989; Böinger et al., 1996) and constantly evolving (Partamies et al., 2019). They usually occur at 100 km altitude and have a horizontal scale size ranging from 10 to 200 km (McEwen et al., 1981; Hosokawa and Ogawa, 2015; Nishimura et al., 2020). The average duration of PsA is around 2 hrs (Jones et al., 2011; Partamies et al., 2017; Bland et al., 2019; Tesema et al., 2020); however, some very long durations (15 hrs) have also been reported (Jones et al., 2013). Pulsating auroras are frequently observed in the nightside equatorward boundary of the auroral oval and during substorm recovery phases in the morning sector. Depending on the level of geomagnetic activity, the time and location of PsA may vary. This variation ranges from observing at all local times during intense geomagnetic activity to being localized to midnight to morning sector around $68°$ of magnetic latitude during weak geomagnetic activity.

Most of the investigations related to pulsating aurora have been multi-measurement case studies such as using the all-sky camera, radars, rocket, riometers, and satellite measurements (Jones et al., 2009; Lessard et al., 2012; McKay et al., 2018; Yang et al., 2019; Nishimura et al., 2020). However, recently a considerable number of statistical findings have been documented, specifically using optical, satellite, incoherent scatter radars, and Super Dual Auroral Radar Network measurements (Jones et al., 2011; Hosokawa and Ogawa, 2015; Partamies et al., 2017; Grono and Donovan, 2018; Bland et al., 2019; Grono and Donovan, 2020; Tesema et al., 2020). It is now well documented that the energies associated with PsA span a wide range from tens to hundreds of keV (Miyoshi et al., 2010, 2015). Pulsating aurora electrons are generally accepted to originate from the magnetosphere near the equatorial plane through pitch angle scattering of energetic electrons into the loss cone by plasma waves (Nishimura et al., 2010, 2011). A source in the magnetospheric equatorial plane implies that pulsating aurora is observed in both hemispheres. However, different shapes and pulsating periods of PsA between hemispheres have also been reported (Watanabe et al., 2007; Sato et al., 2004).

A significant number of studies have used incoherent scatter radars to study the ionization, structure, and energies of precipitating electrons associated with PsA (Wahlund et al., 1989; Böinger et al., 1996; Jones et al., 2009; Hosokawa and Ogawa,





2015; Miyoshi et al., 2015). A recent study by Hosokawa and Ogawa (2015) showed a higher EISCAT electron density at lower altitudes during PsA, which is more pronounced in the morning sector. Similarly, Oyama et al. (2016) showed a maximum electron density below 100 km during a pulsating aurora. Jones et al. (2009) utilized ionization from incoherent scatter radar in Poker Flat, Alaska, to estimate the energy distribution of PsA electrons and compared it with rocket measurements. They showed that the layer of maximum electron density associated with pulsating patches has a thickness of $\sim 15$–$25$ km.

Miyoshi et al. (2015) used EISCAT electron density, Van Allen Probes, and optical data to study the source and the energy of precipitating electrons during PsA. They identified electron density enhancement at altitudes above 68 km associated with the pulsating aurora. Hosokawa and Ogawa (2010) showed a significant ionization in the E region and upper part of the D region (80–95 km) due to energetic precipitation during PsA. This ionization in the D region leads to the appearance of the Pedersen current layer exactly in the altitudes where pulsating ionization occurs and plays a vital role in modifying the current system in

the ionosphere. Hosokawa et al. (2010) used a high time resolution electron density data during PsA and identified enhanced electron density in the E region (95–115 km). They further indicated that the intense ionization could lead to a significant effect on the ionospheric conductivity and current system and, in turn, affect the motion and shapes of PsA patches. Hard precipitation of PsA electrons are known to reach below 70 km and can ionize and change the chemistry of the mesosphere (Turunen et al., 2009, 2016; Tesema et al., 2020). It has also been shown by model results that not all PsA electrons cause strong ionization

and chemical changes (Tesema et al., 2020).

Ionospheric absorption of cosmic radio noise at the D region altitudes has been observed during energetic particle precipitation ($> 10$ keV) associated with PsA (Milan et al., 2008; Grandin et al., 2017; McKay et al., 2018; Bland et al., 2019). Riometric absorption in the ionosphere covers a range of altitudes in the D and E region that contributes to the observed absorption (Wild et al., 2010; Rodger et al., 2012). Thus, observing a one to one correspondence between PsA and ionospheric absorption (Grandin et al., 2017; McKay et al., 2018) further suggests that PsA electrons' energy also cover large ranges. How-

ever, the CNA values are reported to be low during PsA (below $\sim 0.5$ dB), compared to substorm values (above $\sim 1$ dB). These low values suggest that the flux of energetic electrons during substorms are significantly larger than during PsA. HF radio attenuations in the D region from the SuperDARN radars can also be used to detect energetic electron precipitation associated with PsA (Bland et al., 2019).

Based on pulsation, lifetime, and spatial extent, a recent study by Grono and Donovan (2018) categorized pulsating aurora into three groups: patchy, amorphous, and patchy pulsating aurora. Patchy aurora (PA) consists of stable emission structures with pulsations over a limited area of the spatial extent. A patchy pulsating aurora (PPA) is made of steady emission structure with pulsations over a large fraction of their spatial extent, and the amorphous pulsating aurora (APA) is unstable and rapidly varying pulsating aurora. PPA and PA follow the magnetospheric convection, while APA is more dynamic and does not follow

the magnetospheric convection. The occurrence probability of the different types of PsA is reported by Grono and Donovan (2020). The most dominant type is APA, followed by PA and PPA. They reported that before midnight the typical PsA type is APA, while PPA and PA were more common in the late morning. They also estimated the average location of the source regions using T89 model mapping. Before midnight, the source of all types of PsA are constrained in the same area, while after midnight, APA extends further out in the magnetosphere.





An investigation of a few PsA events by Yang et al. (2019) showed a high correlation between CNA absorption and emission intensity of APA type, but no correlation with PPA emission intensity. They also reported the possibility of an extended higher energy range during APA compared to PPA using satellite measurements of a single event. Recently, Tesema et al. (2020) suggested that the abrupt changes in the statistical energy spectrum curve of PsA might be associated with mixing different types of PsA. PsA structure change between patch-like and arc-like and having a characteristic of changing patch size through

time, in general, is suggested to be related to the change in precipitation energy (Partamies et al., 2017, 2019). The question of what are the sources and mechanisms driving different PsAs is still unanswered. A key step to answer this question is to quantify the associated electron fluxes and spectra. In this study, we therefore investigate the altitude and level of ionization, which are related to energies and flux of precipitating electrons, during different PsAs. We use electron density measurements from EISCAT VHF/UHF radars and CNA measurements from the KAIRA riometer. The EISCAT radars and KAIRA measure

the impact from electrons that is truly lost in the atmosphere, compared to incomplete loss cone observations from, for instance, satellite observations. Thus, the height and magnitude of maximum electron density are an indirect measure of energies and flux of precipitating electrons, respectively.

## 2    Materials and Methods

The optical data used in this study are from ground-based all-sky cameras (ASC) operated by Finnish meteorological institute

(FMI). The FMI Magnetometers-Ionospheric Radars-Allsky Cameras Large Experiment (MIRACLE) network consists of 9 ASC located in the Fennoscandian sector. The database has been a huge data source in auroral studies for more than 40 years. As technological advancements were growing and the ASC qualities were degrading in time, two of the digital ASCs with intensified charge-coupled devices (ICCD) were replaced with the newer technology of electron-multiplying CCD (EMCCD) in 2007. Such cameras are more suitable for studying very faint auroral structures (Sangalli et al., 2011), like pulsating aurora,

in detail. For event identification, the entire dataset in this study is from these newer cameras. We use images filtered for the green emission at 557.7 nm in addition to a few events with images of the blue emission at 427.8 nm. From the 9 ASCs, we used Kilpisjärvi (KIL, 69.02°N, 20.87°E, geographic) as our primary data source. However, when there was no data at KIL, a nearby site Abisko (ABK, 68.36°N, 18.82°E, geographic) was used as a substitute. The fields of view (FOVs) of both sites cover a large area around the FOV of EISCAT radars located in Tromsø, as shown in Figure 1.

To study the ionization associated with PsA, we examined electron density measurements from the EISCAT radar located at Tromsø, Norway (69.58°N, 19.21°E, geographic). The EISCAT radar system consists of UHF and VHF radars, which are operating at 931 MHz and 224 MHz frequencies (Rishbeth and Williams, 1985). We used common programme one (CP1), common programme two (CP2) radar modes for UHF, and common programme six (CP6) radar mode VHF radars. These modes are suitable for observing D and E region ionization with a range resolution of < 6 km during particle precipitation

events. Details about the radar modes can be found at https://eiscat.se/scientist/document/experiments/. During the pulsating auroras, either UHF or VHF radar was operative, and electron density was obtained either in the field-aligned or zenith mea-



surements. We then used the magnitude and altitude of electron densities during different types of PsA to understand the flux and energy of electrons associated with them.

We identified 92 pulsating aurora events observed simultaneously by ASC at KIL or ABK and the EISCAT radars at Tromsø
(see Figure 1). The temporal resolutions of optical data was ≤ 10 seconds, and that of the radar data was one minute. This period is significantly longer than the typical period of PsA, which does not allow separation of the on and off phases of PsA. Thus, all the results presented in this study are average statistics over on and off periods of PsA. Types of PsA are identified using keograms and ewograms, as described by (Grono and Donovan, 2018). As these quicklook data formats are not always sufficient and accurate to detect types of PsA, especially during the transition between types, we used individual all-sky camera
images to confirm the detection. APA type can be identified from others by looking at the ewograms and identifying periods, where there are no apparent speed line tracings (i.e. similar structures shown as a blue arrow on the second panel of Figure 2). PPA and PA have more persistent structures, which enable us to identify them quickly. Since PPA has a pulsating nature, striations (alternating bright and dim states of PsA) in the speed line on ewograms are used to differentiate them from PA. During PA, the speed lines have no vertical striations. Once all PsA periods were divided into the three sub-categories, we
then investigated the altitude profile of PsA ionization. The altitude of the maximum electron density and the magnitude of the electron density provides indirect information about the energy and flux of the precipitating PsA electrons, respectively.

Because PsAs cover a wide range of altitudes and electrons energies, comparing the altitude and magnitude of maximum electron density does not always provide the full information on the precipitating electrons. For a detailed investigation of electron density between different PsA types, we average the electron density into five groups with altitude steps of 10 km
between 70 and 120 km. Two of them: between 110 and 120 km and between 100 and 110 km are groups in the E region, and the other three: between 90 and 100 km, between 80 and 90 km, and between 70 and 80 km are groups in the D region of the ionosphere.

We also used measurements of cosmic noise absorption (CNA) made using the Kilpisjärvi Atmospheric Imaging Receiver Array (KAIRA): a radio receiving system located at Kilpisjärvi in Northern Finland (McKay et al., 2015). An observing
frequency of $38.086 \pm 0.098$ MHz was used. Signals from the 48 low-band antennas of KAIRA were cross-correlated with a sample integration of 1 second to form antenna covariance matrices. All-sky radio images can be formed from these using 2D Fourier transforms. However, to achieve the same effect as an optical keogram, only a 1-D Fourier transform of the meridian pixels is made for each time sample, thus forming a "riometric keogram" — or *riogram*. CNA, is determined as $A = 10 \log_{10}(P_{\mathrm{q}}/P)$, where $A$ is absorption in dB, $P$ is the observed power, and $P_q$ is the quiet-sky power derived from a
median of meridian slices from equivalent sidereal times, over a period of 14 days prior to the observation.

The separation between the locations of the FMI camera and KAIRA array is 2.27 km. This proximity means that for observations of ionospheric phenomena in the D region, they have nearly co-incident sky-coverage. Since the field of view of EISCAT lies within the all-sky absorption image, comparing results obtained from EISCAT and KAIRA is also possible. The KAIRA facility has previously been used to study pulsating aurora (Grandin et al., 2017; McKay et al., 2018). The riometry
data corpus from KAIRA spans from 2014 to 2020 and includes 50 events out of the 92 events identified using optical data.



## 3 Results

By inspecting 11 years (between 2010 and 2020) of ASC images from the FMI-MIRACLE network in combination with
EISCAT electron density measurements, we identified PsA events based on classification implemented in Grono and Donovan
(2018) and Yang et al. (2019). In the process, we produced a high time resolution keograms and ewograms (∼10 second

cadence instead of 1 min, as in quicklook data) from ASC images at KIL and ABK. Examples of events that consist of all the
three types of PsAs within a single event are shown in Figures 2, 3, and 4. The panels in these figures from top to bottom show
the keogram, ewograms, EISCAT electron density, and altitude of maximum electron density, and an additional panel with
KAIRA CNA riogram in Figure 4. The red dashed line overlaid in the keo(ewo)grams is the latitude (longitude) of the EISCAT
radar. In Figure 2, APA type is observed between 00:30 and 01:06 UT (green shading), PPA between 01:06 and 01:26 UT (red

shading), and PA between 01:26 and 02:00 UT (purple shading). During these intervals, the single type was dominant over
the FOV of the ASC. However, after 02:00 UT, APA type starts to be apparent in the northwestward direction and lasted until
03:10 UT. After that, the APA type fills the FOV of the camera. For this event, the PsA type between 02:00 and 03:10 UT is
labeled as PA, because the EISCAT radar beam lies within this type of aurora as indicated by red dashed lines. The third panel
in Figure 2 shows the electron density profiles. During the APA type, the electron density ($N_e$) enhancement shows significant

values above 100 km, which is apparent at the beginning and end of this event. In between, we observe PPA and PA types for
which the $N_e$ shows higher values, mostly below 100 km. A corresponding substantial $N_e$ enhancement is observed when the
patchy aurora lies in the FOV of the radar after 1:30 UT. The $N_e$ enhancements during these PsA types reach down to 70 km.
The last panel in Figure 2 shows the altitude of the maximum $N_e$, illustrating a gradual decrease in height at the beginning and
a slight increase at the end of the event. However, equivalent substantial differences in $N_e$ observed below 105 km between

APA and the other two are not captured by the height of maximum electron density.

Figure 3 shows three types of PsA in a single event on January 25, 2012, which lasts more than 6 hrs. The panels displayed
in the figure are the same as in Figure 2. Inspection of this figure shows that different types of PsA are observed in different
regions of the sky. Before 2 UT APA is dominant below 70° latitude followed by a combination of faint unstructured and very
low emission up until 3 UT. After that, PPA becomes dominant below 68° latitude; however, over the EISCAT FOV APA is

dominant until 4 UT. After 4 UT, almost the entire sky is filled with PPA, and then after 5:30 UT, PA starts to appear. The $N_e$
and its maximum altitude in the EISCAT FOV (red dashed lines on keo(ewo)grams) are displayed on the lower two panels of
the figure. There is a clear difference in the magnitude of electron density and the altitude of maximum $N_e$ during the different
types of PsA. The interesting big difference is observed around 4 UT, where there was a transition between APA and PPA. This
transition is also apparent in the altitude of the maximum $N_e$. During APA, the maximum $N_e$ altitude lies at 110 km; however,

during PPA, it is below 100 km. The PA electron density magnitudes and the altitude of the maximum $N_e$ show high variations
corresponding to the patch on and patch off periods over the EISCAT FOV.

Figure 4 shows similar panels as displayed in Figures 2 and 3 with an additional panel of KAIRA CNA riogram with the
EISCAT FOV marked by the red dashed line. In this figure, before 1 UT, the APA type PsA on the top panel is far away from the
EISCAT FOV. However, at nearly 1 UT, a different non-pulsating type of auroral activity, becomes visible over EISCAT. This





is followed by PPA for a very short duration, then up until 2 UT, APA is dominant. Between 2 UT and 3 UT, PPA followed by
PA was observed. The $N_e$ also shows considerable differences during the PsA types. A significant $N_e$ magnitude enhancement
below 80 km around 2:15 UT is seen during PPA. But, such a transition is not apparent in the altitude of the maximum electron
density. Furthermore, the close correspondence between the CNA values and the emission in the keogram is evident. The CNA
values, along with the EISCAT FOV and how deep the ionization occurred, have a nearly one to one correspondence. The CNA

values during PPA are above 1 dB; while during APA, CNA is below 1 dB.

    The thickness of PsA ionization during these three types of PsA showed large differences. Individual electron density profiles
illustrated that PA ionization has a layer thickness of about 20 km, followed by APA with 30 km, and PPA with 40 km thickness.
This is consistently the same in all the three examples displayed in Figures 2, 3, and 4, and from all the electron density profiles
in the study (not shown here). A deeper and higher ionization was observed when the patchy aurora was passing over the FOV

of EISCAT. This is also apparent in the height of the maximum electron density plots with high variations in altitude during
the patch on and off periods.

    As the altitude of the maximum $N_e$ is a single point, it does not reflect the differences in $N_e$ we observe along with the
height profile. For example, in Figure 4, the maximum electron density altitude barely changes, while electron densities during
different PsA types at different altitudes show significant transitions. To include this information in the comparison, we average

electron densities in height bins during different types of PsA. Figure 5 shows histogram of these averages in 10 km intervals
between 70 km and 120 km (panels a–e), as well as a histogram of the maximum altitude electron density (f) for the entire
data set (APA ∼58 hours, PPA ∼ 43 hours, PA ∼ 21 hours). As shown in Figure 5 (a) and (b), there is not much difference
in $N_e$ between APA and PPA types at heights above 100 km. However, PA ionization shows a significant reduction in the
110–120 km region but similar distributions with APA and PPA in the 100–110 km region. In panels (c), (d), and (e), which

corresponds to average $N_e$ between 90 and 100 km, 80 and 90 km, and 70 and 80 km, a substantial shift to the higher $N_e$ is
observed during PPA compared to APA and PA. In the three groups of D region (70–80, 80–90, and 90-100 km altitude ranges),
PPA $N_e$ values were centred around $9.9\,m^{-3}$, $10.7\,m^{-3}$, and $11.3\,m^{-3}$ respectively. However, during APA these values are
$9.3\,m^{-3}$, $10.1\,m^{-3}$, and $10.8\,m^{-3}$. According to Figure 5 (f), precipitation during PPA and PA penetrates deeper on average as
compared to APA. The maximum $N_e$ during PPA and PA primarily lies between below 105 km, while during APA it is above

105 km.

    To further understand the differences between types of PsA, we statistically analyzed the peak electron density and altitude,
as shown in Figure 6. Figure 6 is a two-dimensional histogram, in which the number of points are color coded and shows
the time evolution of the altitude and magnitude of the maximum electron density. Most of the events were observed between
midnight and 9 MLT (07:30 UT), where PPA is more dominant after 5 MLT and no PA before magnetic midnight. The PsA

altitude tends to be lower in the morning sector, especially for PPA and PA. The altitude decrease in the morning sector is
significant in the PPA type, reaching down to 95 km, while the magnitude of the maximum $N_e$ stays above $11.3\,m^{-3}$. The
magnitude of the maximum $N_e$ is higher and more persistent during PPA. However, during PA and APA, the maximum $N_e$
varies a lot with smaller amplitudes (below $11.5\,m^{-3}$). Generally, it is seen that the height of the peak electron density reached
below 100 km during PPA and PA, while during APA, it stays predominantly above 100 km.



The cosmic noise absorption from KAIRA during the three types of PsA is shown in Figure 7. During PPA, the CNA is relatively high compared to the other two types. Based on this figure, the absorption during PPA after 2 MLT shows high values, while the absorption due to APA starts to decline. CNA values during APA are predominantly below 0.5 dB; however, during PPA a substantial number of data points has values above 0.5 dB and reaches values greater than 1 dB. The maximum CNA is observed in the late MLT period (after 5 MLT), which is consistent with the period of high ionization at lower altitudes

observed by the EISCAT radars (see Figure 6). On average, CNA index values during PPA is also higher, 0.33 dB, compared to PA, 0.23 dB, and APA, 0.17 dB as shown by the color coded lines on Figure 7.

## 4    Discussions

The primary purpose of this study is to investigate the differences in fluxes and energies of electrons during different types of pulsating aurora using EISCAT radar electron measurements and KAIRA riometric observations as proxies. Based on keograms

generated from high resolution (∼10 secs) KIL/ABK ASC images, we identified 39 APA, 35 PPA, and 18 PA types with a total of 92 events observed by EISCAT radar. From the collective duration of time, APA was observed for a substantial period of time with 58 hrs, followed by PPA with 43 hrs, and PA with 21 hrs. Grono and Donovan (2020) reported highest probability occurrence of APA and lowest occurrence of PPA using 10 years of ASC data from North America. In our study, PPA was more dominant than PA. The location and precipitation energy of the magnetospheric electrons responsible for the different types of

PsA have been reported to be different (Yang et al., 2019; Grono and Donovan, 2020). The change in the patch sizes during PsAs is also suggested to be an indication of energy deposition in the atmosphere (Partamies et al., 2017, 2019). However, detailed studies about the precipitation energies and the mechanisms behind different structures of PsA are still required. The magnitude of the maximum electron density and its altitude provide both the flux and the energy information about the precipitating PsA electrons. From the electron density measurements, the differences in the magnitude of the electron density and the height of

the maximum electron density among the three types of PsA were significant. The statistical findings presented in this study suggested that PPA has a higher energy range compared to PA and APA types, on average. This is contrary to the results from Yang et al. (2019), which reported that APA has a higher energy range compared to PPA. We also used KAIRA CNA data to further show the differences in energy deposition during the different PsAs. Yang et al. (2019) analyzed CNA from a riometer in Canada to study 12 PsA events (7 APA and 6 PPA) and showed that CNA is systematically higher during APA than during

PPA. They suggested that APA has a higher energy range than PPA by further providing evidence from a single event FAST satellite measurements of electron energy. However, in this statistical study, which has a significant number of events for each PsA type, we found that PPA electrons energies often have a higher energy range than APA, and PA energies lie in between the two. It has been established that CNA (Wild et al., 2010; Rodger et al., 2012) and pulsating aurora (Jones et al., 2009; Partamies et al., 2017) extend over a range of altitudes. Thus, such a contradiction might depend on which altitude the pulsating aurora

and CNA was observed at. It is also possible to find cases where energies of APA electrons are higher than PPA electrons, specifically in the energy range below the energy limit (30 keV) measured by FAST. The stopping altitude of these electrons is above 95 km (Turunen et al., 2009). However, in this study, most of the energy (ionization) differences between the types were





observed below 100 km. This suggests that in higher energy range ($\sim > 30$ keV), PPA electrons' energy is typically higher
than APA. Riometers also respond to precipitating electrons below 30 keV; such particles deposit their energy above 90 km,

which is the D3 region in Figure 6. To further explore the energy difference between PsA types, we use the KAIRA CNA. The
CNA values from KAIRA suggest that the PPA electrons have higher energies to ionize at lower altitudes as compared to APA.
The KAIRA CNA observations showed that the values could reach 1 dB, specifically during PPA types. Such high absorption
values are comparable to values during auroral substorms. However, the average CNA values during APA, PPA, and PA were
0.17 dB, 0.33 dB, 0.23 dB, respectively (Figure 7).

In the late MLT sector (after 5 MLT), PPA tends to be most common with higher electron density at a lower altitude (see
Figure 6). In this study, the occurrence of PA was entirely confined to after 2 MLT. The non-existence of PA before magnetic
midnight is also reported by Grono and Donovan (2020). APA is dominant between 2 MLT and 5 MLT, which is also consistent
with their study. In terms of the order of occurrence, APA tends to be more dominant first, and then PPA or PA follows. Such
an order of occurrence is also reported in Grono and Donovan (2020). As pointed out in their study, it is still unclear if APA is

the onset of the two PsA types. However, this order of occurrence is also an indication that PPA and PA might be associated
with higher energies as compared to APA. In the radiation belt, both the distribution of energetic electrons and chorus waves
activity are dependent on MLT (Aryan et al., 2014; Allison et al., 2017). Allison et al. (2017) showed the persistence of high
flux of electrons with energy $> 30$ keV throughout the dawn sector. On the other hand, Aryan et al. (2014) showed that chorus
wave activity is dominant in the morning to noon period. Thus, the higher electron density observed late in the morning in

this study could be the result of the combination of the two, which include the source and mechanisms for energetic electron
precipitation (For example, see Lam et al., 2010). Oyama et al. (2017) also suggested that the auroral patch formation in the
post-midnight to dawn sector is associated with the development of energetic electron precipitation, despite the low level of
geomagnetic activity. Hosokawa and Ogawa (2015) also reported the descent of the peak electron density associated with an
increase in precipitation energy in the later MLT sector. Their statistical study of 21 PsA events showed that the peak height

of PsA moves below 100 km after 6 MLT. The layer thickness of the categories is also observed to be different. PPA is the
thickest structure with 40 km, while PA is the thinnest structure with 20 km. Thus, the thick ionization layer during PPA is
also associated with precipitating electrons with a broader energy spectrum. This further indicates that the PPA is the most
important PsA type in the D region.

In this study, we found a significant ionization of around 70 km during PPA and PA, which corresponds to 200 keV energy.

Such hard precipitations at this altitude is capable of changing the chemistry of the atmosphere by destroying mesospheric
ozone (Turunen et al., 2016; Tesema et al., 2020). The most likely type of PsA, which contributes largely to such the destruction
of ozone, is PPA (Figure 6). Most of the low fluxes in the higher energy end of PsA spectra observed by Tesema et al. (2020)
might account for APA types. This probable energy spectra difference from satellite measurements among the categories should
be investigated in the future. If exclusively associated with APA, the low flux scenario with no chemical changes in their study

also suggests that it could be possible to visually differentiate which type causes chemical changes and which are not.



## 5 Conclusions

By combining ASC images and EISCAT radar experiments, we identified 39 APA, 35 PPA, and 18 PA types. We used the ionization level to investigate the electron flux and energy range difference between them. The CNA measured by the KAIRA riometer is also used to support the observations. The ionization level during PPA was considerably larger than APA in the region below 100 km. However, the ionization level above 100 km has no significant difference between the two PsA types while PA showed low ionization level. Lower altitudes of the maximum electron density during PPA in the late MLT sector was observed. This suggests that the flux and energy of electrons during PPA is relatively higher than during APA. The CNA from KAIRA was also consistent with the EISCAT electron density results. Higher CNA values during PPA ($\sim > 0.5$ dB) after 3 MLT and lower CNA values during APA ($\sim < 0.5$ dB) after 5 MLT. We also observed a high ionization level down to 70 km in EISCAT measurements, which corresponds to precipitation of relativistic electrons. The mechanisms responsible for the different types of PsA are still unclear, but on average, this study showed that the precipitating electron spectra during the three types of PsA have significant differences, specifically in the higher energy tail. To understand the sources of electrons during different types of pulsating auroras, an ideal combination of measurements would be satellite measurements in the magnetosphere, ground-based optical and radar measurements, and precipitation measurements from a low latitude satellite. Such combinations will be sought in the future.

*Data availability.* The quicklook keograms for event selection are available at https://space.fmi.fi/MIRACLE/ASC/?page=keograms (last access: 28 June 2020). All sky camera data are obtained from FMI-MIRACLE network database, which can be requested from FMI (kirsti.kauristie@fmi.fi). High resolution keograms generated from ASC images and PsA category event list are available at Tesema (2020). The one-minute resolution of EISCAT data used in this analysis is available at http://portal.eiscat.se/schedule/schedule.cgi (last access: 28 June 2020). The interferometric riometry images and keograms are based on cross-correlation statistics data, which are available on request from the KAIRA PI, Sodankylä Geophysical Observatory, http://www.sgo.fi/KAIRA.

*Author contributions.* All authors contribute by providing necessary data, discussions and writing the paper

*Competing interests.* The authors declare that no competing interests are present

*Acknowledgements.* The funding support for F. Tesema and H. Nesse Tyssøy work is provided by the Norwegain Research Council (NRC) under CoE contract 223252. In addition, the work of N. Partamies is supported by NRC project 287427. We thank K. Kauristie, S. Mäkinen, J. Mattanen, A. Ketola, and C.-F. Enell for maintaining MIRACLE camera network and data flow. KAIRA was funded by University of Oulu





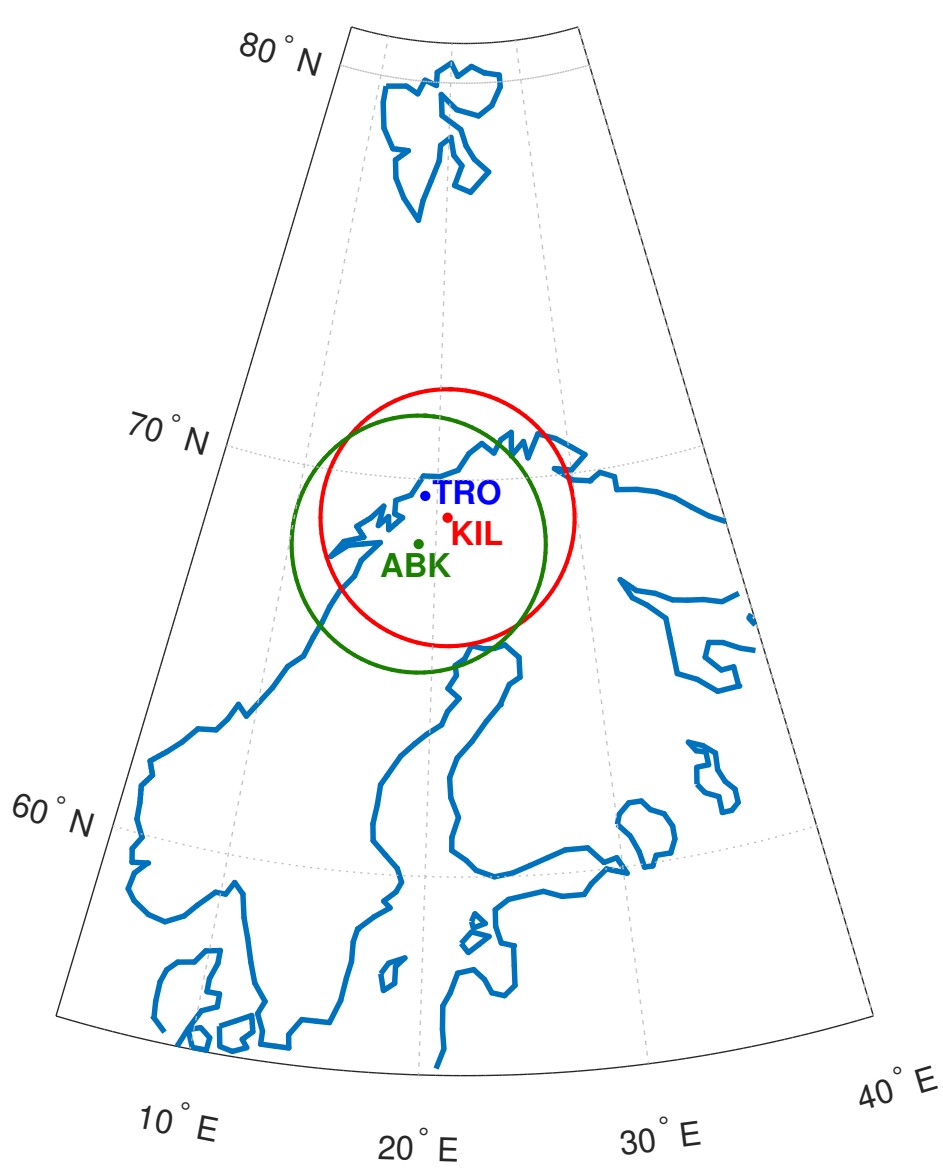

**Figure 1.** Geographic locations of ground-based ASC stations (KIL and ABK) from MIRACLE network and locations of EISCAT radars (TRO).



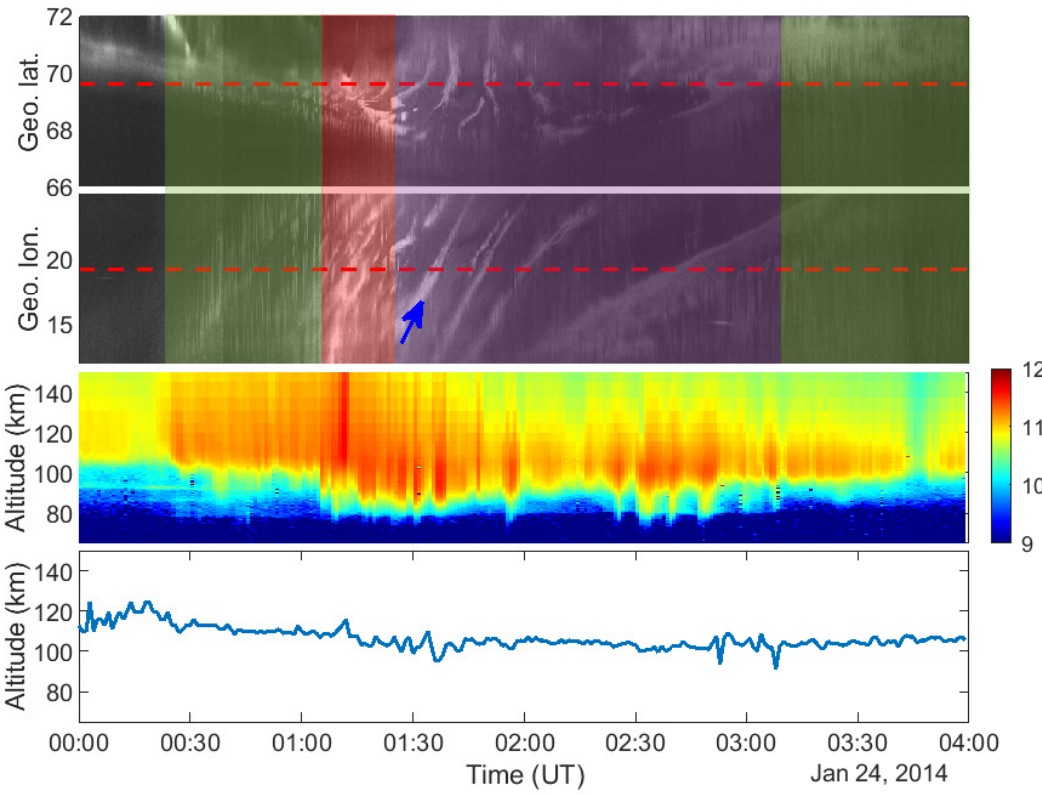

**Figure 2.** Keogram, ewogram, EISCAT electron density, and altitude of maximum electron density on January 24, 2014. The red dashed lines are the latitude and longitude of the EISCAT radars FOV. PsA types: APA (green), PPA (red), and PA (purple) are marked with rectangles in the keo(ewo)grams.

and the FP7 European Regional Development Fund and is operated by Sodankylä Geophysical Observatory with support from the University of Tromsø and volunteer effort.



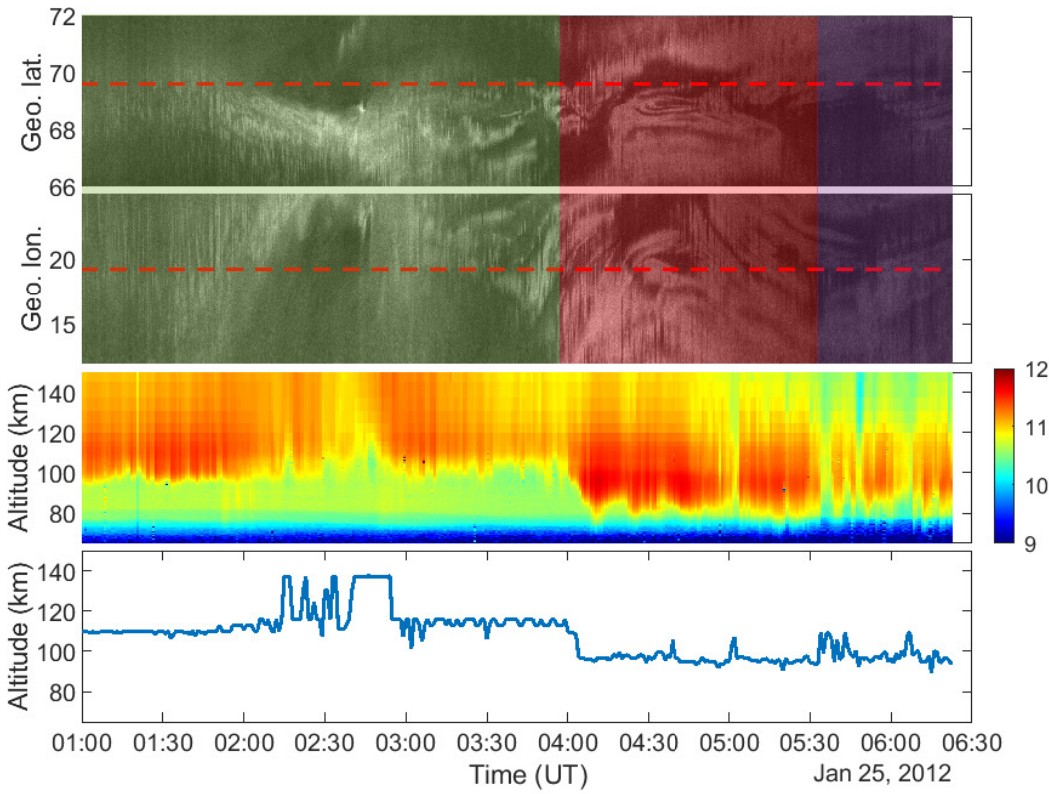

**Figure 3.** Keogram, ewogram, EISCAT electron density, and altitude of maximum electron density on January 25 2012

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




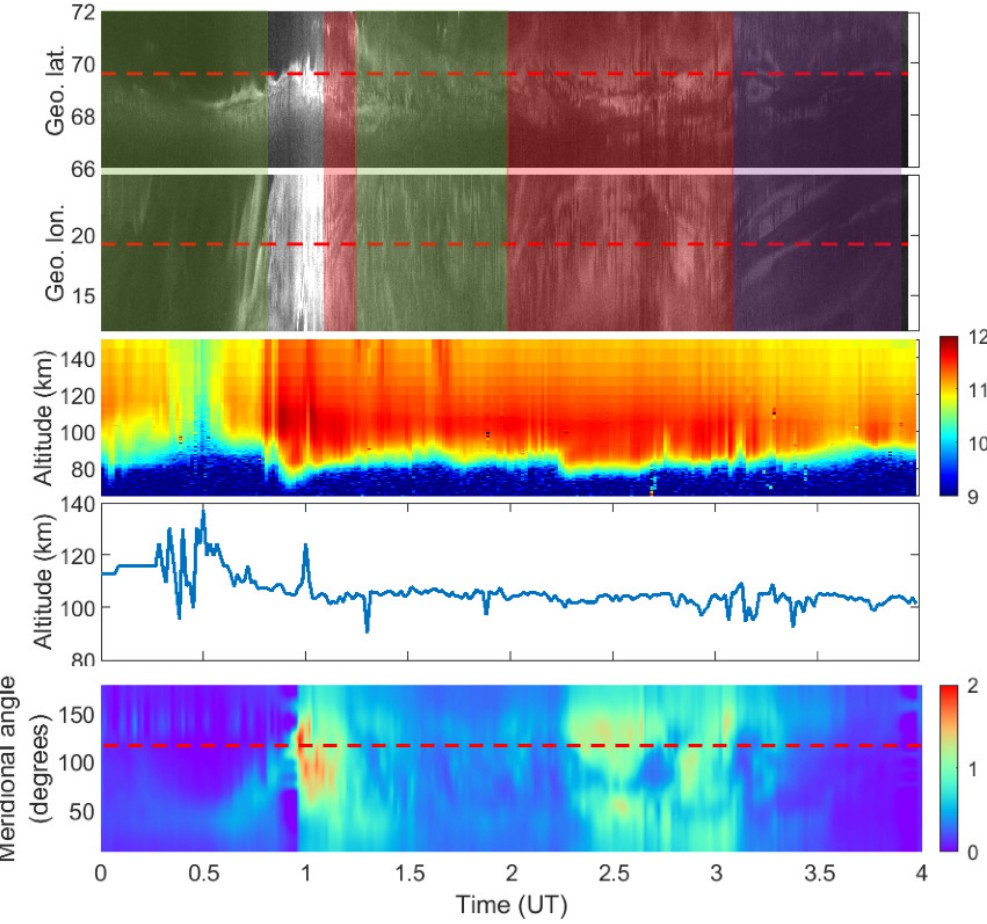

**Figure 4.** Similar to Figure 2 and 3 with additional panel of KAIRA CNA riogram. The red dashed lines overlaid in the riogram is the EISCAT FOV. Note that: the meridional angle in the riogram shows north at 180 degrees and south at 0 degrees, and 90 degrees refer to zenith (location of KAIRA).

Grono, E. and Donovan, E.: Surveying pulsating auroras, Ann. Geophys., 38(1), 1–8, doi:10.5194/angeo-38-1-2020, 2020.

Hosokawa, K., and Ogawa, Y., Ionospheric variation during pulsating aurora, J. Geophys. Res. Space Physics, 120, 5943– 5957, doi:10.1002/2015JA021401, 2015.

Hosokawa, K. and Ogawa, Y.: Pedersen current carried by electrons in auroral D-region, Geophys. Res. Lett., 37(18), doi:10.1029/2010GL044746, 2010.

Hosokawa, K., Ogawa, Y., Kadokura, A., Miyaoka, H. and Sato, N.: Modulation of ionospheric conductance and electric field associated

with pulsating aurora, J. Geophys. Res. Sp. Phys., 115(3), doi:10.1029/2009JA014683, 2010.

Jones, S. L., Lessard, M. R., Fernandes, P. A., Lummerzheim, D., Semeter, J. L., Heinselman, C. J., Lynch, K. A., Michell, R. G., Kintner, P. M., Stenbaek-Nielsen, H. C. and Asamura, K.: PFISR and ROPA observations of pulsating aurora, J. Atmos. Solar-Terrestrial Phys., 71(6–7), 708–716, doi:10.1016/j.jastp.2008.10.004, 2009.



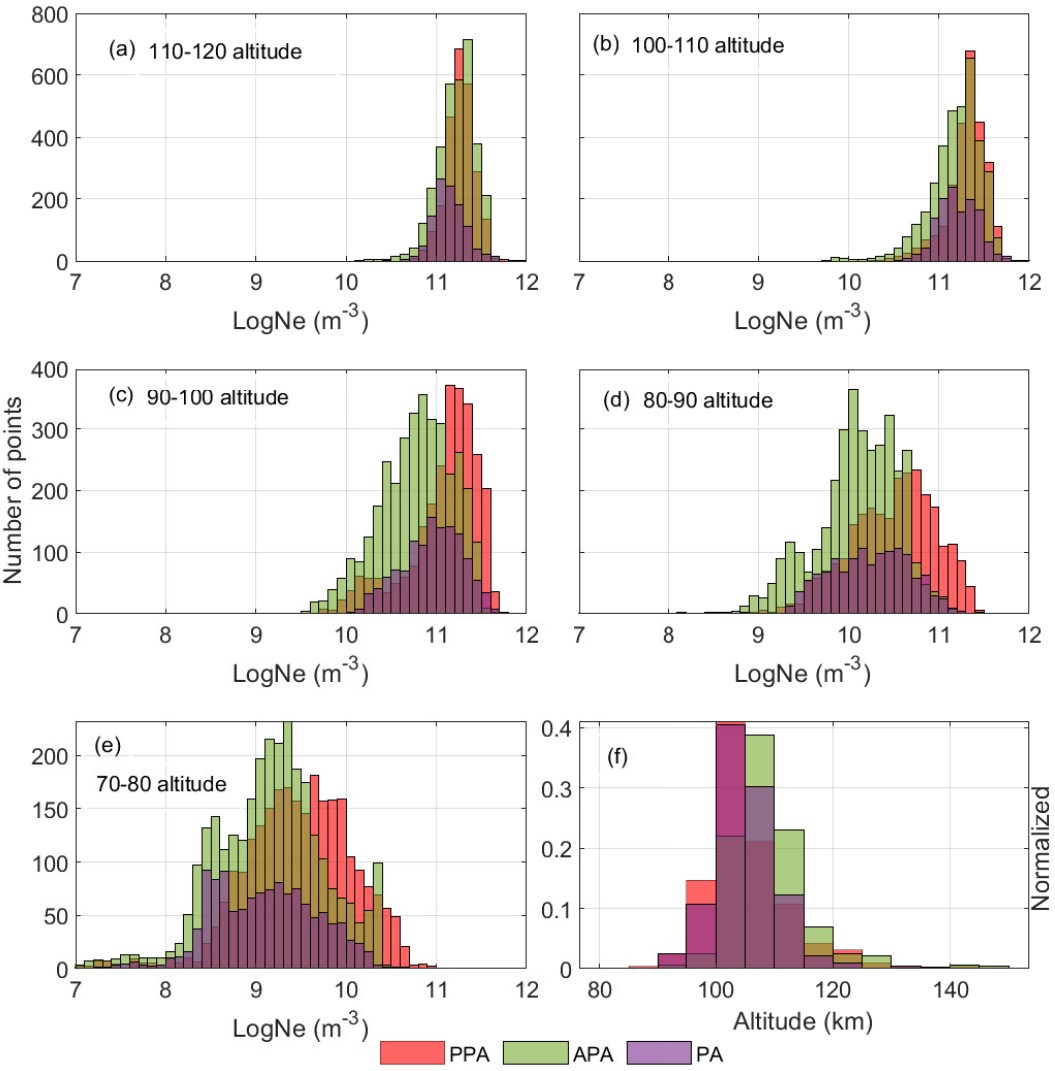

**Figure 5.** Histogram of EISCAT electron density measurements averaged between (a) 110 and 120 km (b) 100 and 110 km (c) 90 and 100 km (d) 80 and 90 km (e) 70 and 80 km and (f) altitude of maximum electron density during different types of pulsating aurora.

Jones, S. L., Lessard, M. R., Rychert, K., Spanswick, E. and Donovan, E.: Large-scale aspects and temporal evolution of pulsating aurora, J.
Geophys. Res. Sp. Phys., 116(3), 1–7, doi:10.1029/2010JA015840, 2011.

Jones, S. L., Lessard, M. R., Rychert, K., Spanswick, E., Donovan, E. and Jaynes, A. N.: Persistent, widespread pulsating aurora: A case study, J. Geophys. Res. Sp. Phys., 118(6), 2998–3006, doi:10.1002/jgra.50301, 2013.

Lam, M. M., Horne, R. B., Meredith, N. P., Glauert, S. A., Moffat-Griffin, T., and Green, J. C., Origin of energetic electron precipitation >30 keV into the atmosphere, J. Geophys. Res., 115, A00F08, doi:10.1029/2009JA014619, 2010.



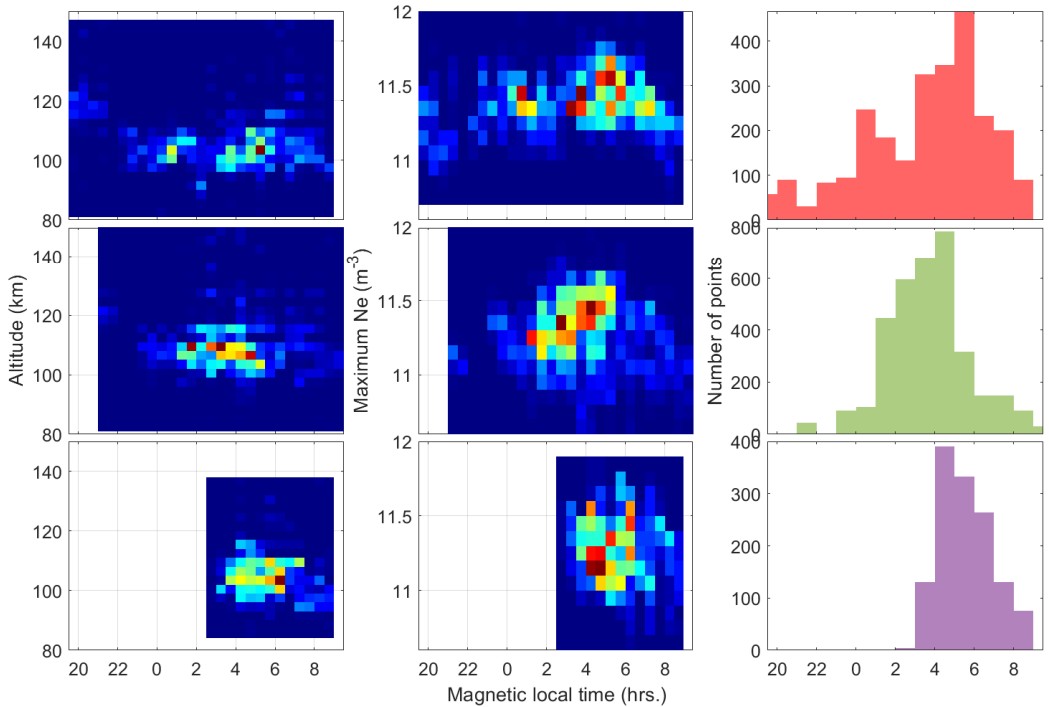

**Figure 6.** 2D histogram of altitude of maximum electron density (left panels) and magnitude of maximum electron density (right panels) during different types of pulsating aurora, PPA, APA , and PA from top to bottom respectively.

Lessard, M. R.: A Review of Pulsating Aurora, Auror. Phenomenol. Magnetos. Process. Earth Other Planets, 55–68, doi:10.1029/2011GM001187, 2012.

McEwen, D. J., Yee, E., Whalen, B. A. and Yau, A. W.: Electron energy measurements in pulsating auroras, Can. J. Phys., 59(8), 1106–1115, doi:10.1139/p81-146, 1981.

McKay-Bukowski, D., Vierinen, J., Virtanen, I. I., Fallows, R., Postila, M., Ulich, T., Wucknitz, O., Brentjens, M., Ebbendorf, N., Enell, C.,
Gerbers, M., Grit, T., Gruppen, P., Kero, A., Iinatti, T., Lehtinen, M., Meulman, H., Norden, M., Orispää, M., Raita, T., de Reijer, J. P., Roininen, L., Schoenmakers, A., Stuurwold, K., and Turunen, E.: KAIRA: The Kilpisjärvi Atmospheric Imaging Receiver Array – System Overview and First Results, IEEE T. Geosci. Remote Sens., 53, 1440–1451, https://doi.org/10.1109/TGRS.2014.2342252, 2015.

McKay, D., Partamies, N. and Vierinen, J.: Pulsating aurora and cosmic noise absorption associated with growth-phase arcs, Ann. Geophys., 36(1), 59–69, doi:10.5194/angeo-36-59-2018, 2018.

Milan, S. E., Hosokawa, K., Lester, M., Sato, N., Yamagishi, H. and Honary, F.: D region HF radar echoes associated with energetic particle precipitation and pulsating aurora, Ann. Geophys., 26(7), 1897–1904, doi:10.5194/angeo-26-1897-2008, 2008.

Miyoshi, Y., Katoh, Y., Nishiyama, T., Sakanoi, T., Asamura, K. and Hirahara, M.: Time of flight analysis of pulsating aurora electrons, considering wave-particle interactions with propagating whistler mode waves, J. Geophys. Res. Sp. Phys., 115(10), 1–7, doi:10.1029/2009JA015127, 2010.



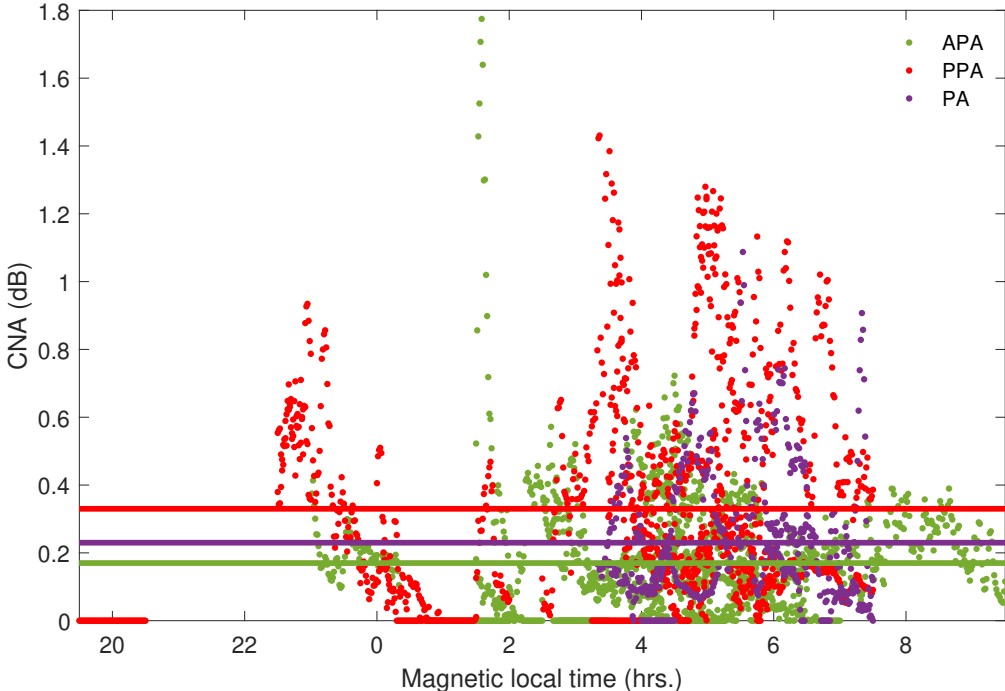

**Figure 7.** KAIRA cosmic noise absorption (CNA) during different types of pulsating aurora. Color coded horizontal lines are the average CNA for respective PsA types.

Miyoshi, Y., Oyama, S., Saito, S., Kurita, S., Fujiwara, H., Kataoka, R., Ebihara, Y., Kletzing, C., Reeves, G., Santolik, O., Clilverd, M., Rodger, C. J., Turunen, E. and Tsuchiya, F.: Energetic electron precipitation associated with pulsating aurora: EISCAT and Van Allen Probe observations, J. Geophys. Res. Sp. Phys., 120(4), 2754–2766, doi:10.1002/2014JA020690, 2015.

Nishimura, Y., Bortnik, J., Li, W., Thorne, R. M., Lyons, L. R., Angelopoulos, V., Mende, S. B., Bonnell, J. W., Le Contel, O., Cully, C., Ergun, R. and Auster, U.: Identifying the driver of pulsating aurora, Sci. (80-. )., 330(6000), 81–84, doi:10.1126/science.1193186, 2010.

Nishimura, Y., Bortnik, J., Li, W., Thorne, R. M., Chen, L., Lyons, L. R., Angelopoulos, V., Mende, S. B., Bonnell, J., Le Contel, O., Cully, C., Ergun, R. and Auster, U.: Multievent study of the correlation between pulsating aurora and whistler mode chorus emissions, J. Geophys. Res. Sp. Phys., 116(11), 1–11, doi:10.1029/2011JA016876, 2011.

Nishimura, Y., Lessard, M. R., Katoh, Y., Miyoshi, Y., Grono, E., Partamies, N., Sivadas, N., Hosokawa, K., Fukizawa, M., Samara, M., Michell, R. G., Kataoka, R., Sakanoi, T., Whiter, D. K., Oyama, S. ichiro, Ogawa, Y. and Kurita, S.: Diffuse and Pulsating Aurora, Space
Sci. Rev., 216(1), doi:10.1007/s11214-019-0629-3, 2020.

Oyama, S.-i., Shiokawa, K., Miyoshi, Y., Hosokawa, K., Watkins, B. J., Kurihara, J., Tsuda, T. T., and Fallen, C. T. ( 2016), Lower thermospheric wind variations in auroral patches during the substorm recovery phase, J. Geophys. Res. Space Physics, 121, 3564– 3577, doi:10.1002/2015JA022129.





Oyama, S., Kero, A., Rodger, C. J., Clilverd, M. A., Miyoshi, Y., Partamies, N., Turunen, E., Raita, T., Verronen, P. T., and Saito, S. , Energetic electron precipitation and auroral morphology at the substorm recovery phase, J. Geophys. Res. Space Physics, 122, 6508–6527, doi:10.1002/2016JA023484, 2017.

Partamies, N., Whiter, D., Kadokura, A., Kauristie, K., Nesse Tyssøy, H., Massetti, S., Stauning, P. and Raita, T.: Occurrence and average behavior of pulsating aurora, J. Geophys. Res. Sp. Phys., 122(5), 5606–5618, doi:10.1002/2017JA024039, 2017.

Partamies, N., Bolmgren, K., Heino, E., Ivchenko, N., Borovsky, J. E. and Sundberg, H.: Patch size evolution during pulsating aurora, J. Geophys. Res. Sp. Phys., 124(6), 2018JA026423, doi:10.1029/2018JA026423, 2019.

Rishbeth, H., and P. J. S. Williams, The EISCAT ionospheric radar: The system and its early results, Q. J. R. Astron. Soc., 26, 478–512, 1985.

Rodger, C. J., Clilverd, M. A., Kavanagh, A. J., Watt, C. E. J., Verronen, P. T., and Raita, T., Contrasting the responses of three different ground-based instruments to energetic electron precipitation, Radio Sci., 47, RS2021, doi:10.1029/2011RS004971, 2012.

Royrvik. O., Davis, T. N.: Pulsating aurora local and global morphology, , 82(29), 4720–4740, 1977.

Sangalli, L., Partamies, N., Syrjäsuo, M., Enell, C. F., Kauristie, K. and Mäkinen, S.: Performance study of the new EMCCD-based all-sky cameras for auroral imaging, Int. J. Remote Sens., 32(11), 2987–3003, doi:10.1080/01431161.2010.541505, 2011.

Sato, N., Wright, D. M., Carlson, C. W., Ebihara, Y., Sato, M., Saemundsson, T., Milan, S. E. and Lester, M.: Generation region of pulsating aurora obtained simultaneously by the FAST satellite and a Syowa-Iceland conjugate pair of observatories, J. Geophys. Res. Sp. Phys., 109(A10), 1–15, doi:10.1029/2004JA010419, 2004.

Tesema, F., Partamies, N., Tyssøy, H. N., Kero, A. and Smith-Johnsen, C.: Observations of electron precipitation during pulsating aurora and its chemical impact, J. Geophys. Res. Sp. Phys., n/a(n/a), e2019JA027713, doi:10.1029/2019JA027713, 2020.

Tesema, Fasil: Replication data for: Observations of precipitation energies during different types of pulsating aurora. figshare. Dataset. doi:10.6084/m9.figshare.12559127, 2020

Turunen, E., Verronen, P. T., Seppälä, A., Rodger, C. J., Clilverd, M. A., Tamminen, J., Enell, C. F. and Ulich, T.: Impact of different energies of precipitating particles on NOx generation in the middle and upper atmosphere during geomagnetic storms, J. Atmos. Solar-Terrestrial Phys., 71(10–11), 1176–1189, doi:10.1016/j.jastp.2008.07.005, 2009.

Turunen, E., Kero, A., Verronen, P. T., Miyoshi, Y., Oyama, S. I. and Saito, S.: Mesospheric ozone destruction by high-energy electron precipitation associated with pulsating aurora, J. Geophys. Res., 121(19), 11852–11861, doi:10.1002/2016JD025015, 2016.

Wahlund, J. E., Opgenoorth, H. J. and Rothwell, P.: Observations of thin auroral ionization layers by EISCAT in connection with pulsating aurora, J. Geophys. Res. Sp. Phys., 94(A12), 17223–17233, doi:10.1029/JA094iA12p17223, 1989.

Watanabe, M., Kadokura, A., Sato, N. and Saemundsson, T.: Absence of geomagnetic conjugacy in pulsating auroras, Geophys. Res. Lett., 34(15), doi:10.1029/2007GL030469, 2007.

Wild, P., Honary, F., Kavanagh, A. J., and Senior, A. , Triangulating the height of cosmic noise absorption: A method for estimating the characteristic energy of precipitating electrons, J. Geophys. Res., 115, A12326, doi:10.1029/2010JA015766, 2010.

Yamamoto, T.: On the temporal fluctuations of pulsating auroral luminosity, J. Geophys. Res., 93(A2), 897, doi:10.1029/JA093iA02p00897, 1988.

Yang, B., Spanswick, E., Liang, J., Grono, E. and Donovan, E.: Responses of Different Types of Pulsating Aurora in Cosmic Noise Absorption, Geophys. Res. Lett., (1961), 5717–5724, doi:10.1029/2019GL083289, 2019.