# Peer review of "Observations of precipitation energies during different types of pulsating aurora"

_Annales Geophysicae, 2020_

## Referee Comment (RC1) · Anonymous Referee #1 · 17 Jul 2020

The authors investigate precipitating electron energies during different types of pulsating aurora (PsA). Via statistically analyses of data measured with all-sky image cameras, EISCAT radars and KAIRA, they report that altitude profiles of electron density and cosmic noise absorption values vary depending on types of pulsating aurora: amorphous pulsating aurora (APA), patchy pulsating aurora (PPA), and patch aurora (PA). These results are important and deserve publication for improving understanding of what mechanisms create differences in the shape of pulsating aurora. However, I have a few comments for the authors to consider a minor revision before the publication.

Specific comments

1. The difference between PPA and PA

   - In lines 132-133, it is described that PPA has a pulsation nature and PA does not. It seems inconsistent with the description that both PA and PPA have steady emission structures with pulsations in lines 81-83.

2. The division of PsAs into the three sub-categories

   - PA, PPA and APA are often alternately observed in the EISCAT FOV during a short period. How fine are you classifying them? Is the categorization process quantitative to eliminate arbitrariness?
   - The auroras classified into PA in Figure 2 have an arc-like shape not patch shape. I checked 557.7-nm all-sky images installed in Tromsø. Especially, the aurora indicated by the blue arrow in Figure 2 has vorticities. Isn't there a possibility that they are discrete auroras or arc-type diffuse aurora not patchy auroras?

3. The definition of PsA thickness

   - In line 197, the thickness of PsA is mentioned. What is the definition of thickness? Is it the full width at half maximum or anything else?

4. The meridional angle in the riogram

   - I suggest that the vertical axis of the bottom panel in Figure 4 should be converted to the geographic latitude from the meridional angle to make it easier to compare with top two keograms.

5. The KAIRA cosmic noise absorption

   - In lines 228-229, it is described that "The maximum CNA is observed in the late MLT period (after 5MLT)", but the maximum CNA during APA is observed in before 2 MLT and that during PPA is observed before 4 MLT.

[Figure]

6. The MLT distribution of electron densities during different types of PsA

   - In lines 265-266, it is described that "In the late MLT sector (after 5MLT), PPA tends to be most common with higher electron density at a lower altitude (see Figure 6).", but it seems that PA is also common until 7 MLT in Figure 6.

Technical corrections

1. line 2, "few" should be "a few"

2. line 23, "the" should be "a"

3. liens 27-28, the sentence starting with "In general ..." needs a reference.

4. line 29, the sentence starting with "The auroras are ..." is incomplete and needs to be revised.

5. lien 30: "PsA" should be "PsAs"

6. line 73: "region" should be "regions"

7. line 106: "ASC" should be "ASCs"

8. line 119: "region" should be "regions"

9. line 212: "Ne" should be "log10(Ne)"

10. lines 221, 233: The unit symbols should be in roman type

11. line 260: "D3" should be "D"?

12. Figure 4: The date should be specified.

13. Figure 5: The color bar should be shown.

---

## Referee Comment (RC2) · Anonymous Referee #2 · 9 Aug 2020

Tesema et al. used data from all-sky Cameras, EISCAT radars and KAIRA to investigate precipitating electron energies during different types of pulsating aurora and found different statistical altitude profiles of electron density and cosmic noise absorption for different types of aurora. The results are important for further understanding of the origins of electrons responsible for each type of pulsating aurora. However, the categorization process in this study needs more clarification as it is important for the following statistical analysis. More detailed comments are presented below.

Major comments:

1. The FoV of EISCAT is pretty small compared to ASCs. And PPA and APA are sometimes hard to be distinguished from keogram and ewograms. For example, the first shading area in the ewogram of Figure 2 seems like a mixture of PPA and APA.

[Figure]

How reliable is your categorization process? It would be helpful if a few movies of ASC images with FoVs of EISCAT are provided.

2. The red shading area in the keogram of Figure 2, the auroral structure in the EISCAT FoV seems to be more like auroral rays or streamers to me. Though there are pulsating aurora at lower latitudes. Is it possible some of your events are discrete auroras other than pulsating auroras?

3. PPA, APA and PA may be alternately presented in the EISCAT FoV in a short time period. How fine are you classifying them?

4. What is the beam size of KAIRA around the FoV of EISCAT? It's better to present FoVs of KAIRA beams in Figure 1 as well. Is it possible there are different types of pulsating aurora in the beam?

Minor comments:

1. Lines 81-83: The definition of PA and PPA seems the same to me here and is inconsistent with lines 132-133.

2. Line 128: ewogram – from which latitude is the ewogram constructed? Please clarify.

3. Figure 4: It's better to change the vertical axis into Glat, so it's easy for readers to compare the riogram with keograms.

---

## Author Comment (AC1) · 16 Sep 2020

We thank the referee for evaluating the manuscript and forward constructive comments. We include corrections and suggestion made by the reviewer by adding texts, references and modifying figures. Point by point responses to the reviewer comments are listed below.

Specific comments

1. The difference between PPA and PA

- In lines 132-133, it is described that PPA has a pulsation nature and PA does not. It seems inconsistent with the description that both PA and PPA have steady emission structures with pulsations in lines 81-83.

[Figure]

The main difference between the two types is the spatial extent of the pulsation, PPA has stable structure and pulsating over a large area but PA has a limited area pulsation. The patch outlines/shapes are stable over several pulsations for PPA, unlike APA, for which the structures are too transient to be tracked. To describe this difference we added a text (line 134)

2. The division of PsAs into the three sub-categories

- PA, PPA and APA are often alternately observed in the EISCAT FOV during a short period. How fine are you classifying them? Is the categorization process quantitative to eliminate arbitrariness?

Classifying PsA is entirely based on visual inspection of keograms and whenever we find it difficult based on keograms we flip through the ASC images to make the classification more reliable. However, the time resolution of EISCAT measurement we use here is 1 minute, so any transient structures less than this time cannot be discussed. We include data for PsA types which is dominant for at least 10 minutes and based on dominating over other type of aurora.

- The auroras classified into PA in Figure 2 have an arc-like shape not patch shape. I checked 557.7-nm all-sky images installed in Tromsø. Especially, the aurora indicated by the blue arrow in Figure 2 has vorticities. Isn't there a possibility that they are discrete auroras or arc-type diffuse aurora not patchy auroras?

We probably included some transient structures in the data, however, EISCAT data time resolution we used is 1 minute, such transient structures cannot be captured. For further reference we add movies of PsA types of Figure 2. In addition, a patch can be an elongated one, so an arc like diffuse aurora does not change the precipitation as much as the discrete aurora does.

3. The definition of PsA thickness

- In line 197, the thickness of PsA is mentioned. What is the definition of thickness? Is
it the full width at half maximum or anything else?

Yes, it is full width at half maximum and text added to clarify it (line 199).

4. The meridional angle in the riogram

- I suggest that the vertical axis of the bottom panel in Figure 4 should be converted to the geographic latitude from the meridional angle to make it easier to compare with top two keograms.

Vertical axis labels changed.

5. The KAIRA cosmic noise absorption

- In lines 228-229, it is described that "The maximum CNA is observed in the late MLT period (after 5MLT)", but the maximum CNA during APA is observed in before 2 MLT and that during PPA is observed before 4 MLT.

Corrected (line 230): "Most of the high CNA values are observed during the late MLT period (after 3 MLT)"

6. The MLT distribution of electron densities during different types of PsA - In lines265-266,it is described that"In the late MLT sector (after 5MLT),PPA tends to be most common with higher electron density at a lower altitude (see Figure 6).", but it seems that PA is also common until 7 MLT in Figure 6.

Corrected (line 267): "In the late MLT sector (after 5 MLT), PPA and PA are more common with higher electron density at a lower altitude (see Figure 6)"

Technical corrections

1. line 2, "few" should be "a few"

2. line 23, "the" should be "a"

3. liens 27-28, the sentence starting with "In general ..." needs a reference.

4. line 29, the sentence starting with "The auroras are ..." is incomplete and needs to be revised.

5. lien 30: "PsA" should be "PsAs"

6. line 73: "region" should be "regions"

7. line 106: "ASC" should be "ASCs"

8. line 119: "region" should be "regions"

9. line 212: "Ne" should be "log10(Ne)"

10. lines 221, 233: The unit symbols should be in roman type

11. line 260: "D3" should be "D"?

12. Figure 4: The date should be specified.

All the above 12 points are corrected as suggested.

13. Figure 5: The color bar should be shown. It is shown below the x axes labels of the last two panels.

Please also note the supplement to this comment:
https://angeo.copernicus.org/preprints/angeo-2020-43/angeo-2020-43-AC1-supplement.zip

---

## Author Comment (AC2) · 16 Sep 2020

We thank the referee for evaluating the manuscript and forward constructive comments. We include corrections and suggestion by adding texts, references and modifying figures. Point by point responses to the reviewer comments are listed below.

Major comments:

1.The FoV of EISCAT is pretty small compared to ASCs. And PPA and APA are sometimes hard to be distinguished from keogram and ewograms. For example, the first shading area in the ewogram of Figure 2 seems like a mixture of PPA and APA. How reliable is your categorization process? It would be helpful if a few movies of ASC images with FoVs of EISCAT are provided.

[Figure]

All the classification is based on EISCAT-FOV location in ASC images, and yes it is quite small but the pointing direction is well known and patchy features are typically large. The better approach often was flipping through all ASC images and selecting the dominant one over EISCAT. Movies for Figure 2 is provided as an example (supplement). As the sub-classification has been done twice (once at the beginning of the project and once when finalizing the analysis), we are confident on achieving a sufficient accuracy.

2.The red shading area in the keogram of Figure 2, the auroral structure in the EISCAT FoV seems to be more like auroral rays or streamers to me. Though there are pulsating aurora at lower latitudes. Is it possible some of your events are discrete auroras other than pulsating auroras?

We shaded the whole area on Figure 2,3 and 4, to show the different types of PsA over the entire ASC FOV and we mainly use the EISCAT data when a dominant PsA type is observed over EISCAT and stayed for few minutes (>10 minutes) and based on dominating over other type of aurora.

3.PPA, APA and PA may be alternately presented in the EISCAT FoV in a short time period. How fine are you classifying them?

We use visual inspection as the main way to classify them and exclude durations when we are not sure of the types. We include the dominant PsA types which persists a relatively longer period (> 10 minutes) (text added on line 136)

4.What is the beam size of KAIRA around the FoV of EISCAT? It's better to present FoVs of KAIRA beams in Figure 1 as well. Is it possible there are different types of pulsating aurora in the beam?

The KAIRA data used here is based on the riometric imaging where the images do not consist of individual beams any longer. Yes, the KAIRA spatial resolution 24km at 90 km altitude (added on line 149) is relatively large and might also include different
types of PsA. However, we used CNA values over EISCAT FOV corresponding to the dominant PsA types to discuss the absorption differences displayed on Figure 7.

Minor comments:

1.Lines 81-83: The definition of PA and PPA seems the same to me here and is inconsistent with lines 132-133.

The main difference between the two types is the spatial extent of the pulsation, PPA has stable structure and pulsating over a large area but PA has a limited area pulsation. . The patch outlines/shapes are stable over several pulsations for PPA, unlike APA, for which the structures are too transient to be tracked. To describe this difference we added a text (line 134).

2.Line 128: ewogram – from which latitude is the ewogram constructed? Please clarify.

Corrected (line 129): "Types of PsA are identified using keograms and ewograms generated at the location of the ASC, as described by . . . "

3.Figure 4: It's better to change the vertical axis into Glat, so it's easy for readers to compare the riogram with keograms

Vertical axis lable corrected.
* * *

---

## Author Response (AR1)

**Response to the editor's comments**

Comments are in black and responses in red
* * *
These seem like "raw" all-sky images, in which case the latitude and longitude scales should not be linear - pixels near the edge cover more degrees of latitude/longitude than pixels near the centre, for the same height aurora. Could you also double check the latitude and longitude scales on the figures in the paper and confirm they are calculated correctly. If not, then please also confirm that the mistake does not affect your conclusions through a mis-calculation of the position of the EISCAT and KAIRA observations.

Yes, the movies are generated from raw images. Since, KAIRA and EISCAT locations are quite near to the KIL-ASC location, we use a linear scale, and it is good enough for the data analysis and results presented in the paper. We confirm that all latitudes and longitudes appeared on figures are correct and no effect that can affect the conclusion. We include the movies to show further how different types of PsA looks like from the raw ASC images and over the EISCAT FOV.

I suggest also making sure the time labels in the videos have two digits for each of hours, minutes, seconds, i.e. including the leading/trailing zero, for clarity, and also adding labels to the axes.

Labels added and time stamp corrected.